# Study on Passive Heating Involving Firewalls with an Additional Sunlight Room in Rural Residential Buildings

**DOI:** 10.3390/ijerph182111147

**Published:** 2021-10-23

**Authors:** Simin Yang, Bart Dewancker, Shuo Chen

**Affiliations:** Faculty of Environmental Engineering, The University of Kitakyushu, Kitakyushu 808-0135, Japan; bart@kitakyu-u.ac.jp (B.D.); chensure8718@163.com (S.C.)

**Keywords:** firewall–additional sunlight system, passive heating, rural residential buildings, southern Shaanxi, software analysis

## Abstract

With the growth in China’s economic GDP, energy consumption has increased year by year. The energy demand of rural residential buildings is 223 million tons of standard coal equivalent, accounting for 24% of the national energy demand. Therefore, an energy-saving design for rural residences is necessary. This research took the traditional residences in southern Shaanxi as the research object and combined the cooking methods in southern Shaanxi with solar heating, proposing a sunlight heating system with an additional firewall. The system is composed of a firewall system and a sunlight system. The combination of the two systems prolongs the heating time and makes up for the lack of intermittent heating. The firewall principle involves using the heat generated by cooking through the heat storage and heat release capacity of the wall, and using the principle of heat radiation and convection to increase the indoor temperature. Meanwhile, the principle of the additional sunlight room involves using the external facade of the building to establish an additional sunlight room, by absorbing the heat radiation of the sun and using the principle of heat transfer from the wall. The rapid loss of indoor hot air is avoided, the heating time is prolonged, and part of the heat is retained, thereby improving the heating efficiency. A model was established based on the typical residential model in southern Shaanxi, and the presence or absence of solar radiation on the wall was used as the research variable. Using ANSYS software to simulate the analysis, it is concluded that the firewall–sunlight system can extend the heating time and meet the continuous heating demand, and the heating effect is better than that of the firewall heating system alone. When the walls have solar radiation, the annual heat load reduction rate of the buildings under the new system is 20.21%. When the walls do not have solar radiation, the annual heat load reduction rate of the buildings under the new system is 8.56%.

## 1. Introduction

### 1.1. Motivation

Nowadays, with the rapid growth in the global economic GDP, the demand for energy in construction, agriculture, industry, and other industries has increased [1]. From 2016 to 2018, the construction industry accounted for 12.5% of the global energy requirement, increasing to 21.22% [2]. Therefore, the demand for energy in the construction industry is on the rise. The total annual energy demand of the building industry was 2.147 billion TCE (tons of standard coal equivalent—1 ton standard coal equivalent is the conversion index for calculating various energy quantities according to the calorific value of standard coal), accounting for 46.5% of the whole energy requirement in 2018. The total carbon emissions generated by the construction industry is 4.93 billion TCE, accounting for 51.3% of the national carbon emissions [3]. It is expected that in 2050, this value will double [4]. According to China’s construction Industry Energy Report 2016, residential buildings account for 77% of building energy consumption [5]. Thus, residential buildings are an important part of building energy consumption. In response to this phenomenon, some experts have proposed the use of renewable sources, such as geothermal and hydroelectric power generation, to save energy [6]. Neupane et al. [7] evaluated the development space of wind energy in provincial areas of Nepal and established an energy model based on the local geographical environment, economic conditions, and other factors. Through the calculation of the open source system platform, 1686 MW of local wind energy can be generated. In addition, the local area has the potential to generate 267.0 MW of wind power, and the construction cost is 46 USD/MWh. The development of wind energy has a good impact on resource utilization in Nepal. Yuan et al. [8] found, through high-density instruments, that there is a high-temperature underground heat storage layer deep underground in the Yangbajing area. When using five-point configuration with multiple wellheads, the flow rate of a single geothermal heat generation is 17 kg/s. The total capacity of geothermal power can reach 66.0 MW, which can effectively reduce greenhouse gas emissions. Rauf et al. [9] designed a combination of solar photovoltaic and hydropower, analyzing the impact of solar photovoltaic on hydropower. Using a combination of the two methods, the total power of the hydropower generation increased by 3.5%.

In China, to effectively reduce the energy demand of residences, the housing and urban–rural construction departments have proposed a series of urban residential energy-saving design specifications and have achieved some results [10]. Many scholars have put forward corresponding urban residential building energy-saving design strategies through a variety of research methods [11]. However, there are few considerations for the energy-saving design of rural residences. Urban houses are different from rural houses in terms of architectural design, climatic conditions, and behavior of occupants [12]. In terms of architectural design, in order to reduce energy demand, urban residential buildings generally adopt centralized high-rise buildings. High-rise residential buildings are divided according to the internal space, and there are two types: the corridor type and the unit type. Its characteristics are a long service life and a large demand [13]. Rural housing is generally self-built, with the residents getting involved in the construction mainly based on their years of construction experience. Therefore, residential houses not only have the function of use, but also have the function of cultural inheritance [14]. In terms of climatic conditions, urban residences are greatly affected by solar radiation, and electrical appliances are mainly used for indoor adjustment [15]. The indoor temperature of rural houses in the summer is mainly regulated by natural ventilation, as well as partly by electrical equipment. Heating in winter mainly relies on burning charcoal fires [16]. In terms of occupant behavior, compared to rural residents, urban residents pay more attention to privacy. Therefore, in addition to satisfying the basic space, urban housing has less public communication space [17]. Residents in rural areas live on cultivated land and need space to store agricultural tools and crops. In addition, there is a lot of communication between residents, and these behavioral factors make rural residential building spaces larger, relatively open, and possessing other characteristics [18]. In the light of the statistics from the National Bureau of Statistics, the per capita housing area of rural residents in 2019 was 48.9 m^2^, an improvement of 97.2% compared to 2000 [19]. Therefore, energy-saving designs for rural houses is very necessary. Rural residential buildings have the advantages of being relatively open, flexible in layout, and large in space [20]. Thus, when studying the energy efficiency of rural residences, the reconstruction of buildings and the utilization of renewable energy have great advantages [21]. In this study, passive cooking and the addition of a sunlight room heating system are proposed for winter, which can effectively improve the indoor temperature and reduce energy requirements.

### 1.2. Literature Studies

In rural areas, energy consumption mainly comes from residential buildings, and nearly one-fifth of the energy demand is used for the heating and cooling of residences [22]. With the increase in energy consumption, environmental pollution and other problems also follow [23]. In China, the demand for the energy consumption of residential buildings in rural areas is even more severe [24]. To solve this problem, relevant departments have made policies to reduce the demand for energy [25]. Wang et al. [26] studied the factors affecting Chinese household energy consumption (HEC) from 2005 to 2017, and explored the difference between HEC in urban and rural areas. The HEC is higher in rural areas, and it is recommended to decrease the demand for energy. Satish et al. [27] established a per capita energy balance for urban and rural households by analyzing the energy transition and requirement patterns in India. They divided the income into three types—high, medium, and low—and analyzed the relationship between energy and income, finding that cooking is one of the main energy demands. These results explain the difference between the urban and rural requirements of each energy option in India. Zi et al. [28] investigated the possibility of HEC and the purchase of energy-consuming products by analyzing the policy of clean energy transition in rural areas of Henan Province. It is believed that with the development of the economy, rural areas should implement an energy transition policy, and it is very necessary to increase the supply of clean energy.

At present, the research on passive energy saving in buildings mainly includes three aspects: the first approach is to change the maintenance structure of the phase-change materials [29,30], the second is to effectively use renewable energy [31], and the third is to burn biomass [32]. Maryam et al. [29] studied the thermal properties of external walls in Morocco and compared and analyzed the thermal properties of different materials through the calculation of the four-bar method. The research results showed that the selection of hemp and cotton material can effectively reduce greenhouse gas emissions, with an average annual reduction of 30%. Ahmed et al. [33] built a mathematical model to calculate the heat gain value, economic cost, and sunshine conditions of different materials. By optimizing the objective function, the heat gain value of a building can be reduced by 34% and the daylighting situation can be increased by 11%, thereby effectively improving energy utilization. Ji et al. [34] believed that biochar can improve the damp and heat performance of buildings through the comparison of multiple groups of experiments. The thermal conductivity of biochar–mortar can be reduced by 57.6% and the water vapor resistance coefficient increased by 50.9% compared to the single mortar. The research results showed that the mortar exterior wall with biochar can improve the strength, heat, and humidity performance of mortar.

Facing the problem of increasing energy consumption in residential buildings, some experts believe that the effective use of solar energy has potential [35]. For nearly a decade, solar energy has been used to design energy-efficient residential buildings [36]. The research on the effective use of solar energy mainly includes the following two points: The first is to study the types of solar energy storage, such as glass boxes [37], Tromble walls [38], and solar spaces [39]. The second uses solar energy to reduce the energy consumption of buildings [40]. Hilliaho et al. [37] studied 156 different types of glass balconies in Finnish apartments in the 1970s. Five important factors affecting the design of glass balconies were summarized, including the airtightness of glass, the heat absorption coefficient, heat loss, and the building ventilation system. Simões et al. [41] combined solar energy with Trombe walls based on the Mediterranean climate type and used software to compute the energy performance of the system. The new system can reduce heating demand by 20%. Duan et al. [42] put phase-change materials into Tromble walls and compared the best combination of phase-change materials and Tromble walls in Beijing, Jiuquan, and Shenyang. The melting temperature and layer thickness of the material were 23 °C and 5 mm (Beijing, Jiuquan) and 21 °C and 5 mm (Shenyang), respectively. Li et al. [43] took cold rural areas as an example and compared solar space with solar space containing PCM shutters. The results showed that the solar space heating effect with PCM blinds is obvious, saving 5.27% energy. Barrencua et al. [44] combined solar space with a mechanical ventilation system, combined with software calculation and an experiment, and showed that the energy saving of this system is 38.48 kWh·m^−2^, with 58% in heating. Using solar energy has been proven to decrease the energy consumption of buildings. Mihalakakou et al. [45] conducted simulation calculations by setting different parameter values of the solar heating system to simulate the thermal performance and thermal environment of the winter and summer seasons. The results showed that a solar heating system is an effective heating system in cold seasons. Mottard et al. [46] studied an additional solar energy system and demonstrated the reliability of the model by comparing the calculation with the measured experimental data through the mathematical model. The additional solar energy system is effective at saving energy.

Satisfying the temperature of the human body by burning the heat generated by biomass is also a passive heating method [47]. This heating mode first appeared in the spring and autumn periods of China and has been handed down to this day, which is called a kang [48]. As of 2014, the main method in China for 174 million people in winter is heating on a kang [49,50]. A kang is mainly made of three parts: a body, stove, and chimney [51]. Stoves are generally used for burning firewood and cooking food [52]. In addition to heating food, the heat generated by burning biomass also heats the body through its inner cavity [53]. The hot smoke generated therein is then discharged to the outdoors by the chimney [54]. Heated kang is the use of the body to heat the local temperature of the human body, to meet the temperature of the human body rather than raise the temperature of the whole room [55]. The research on the heating performance of a kang can be divided into two aspects: the first is to put forward the reconstruction scheme of the body [56,57]; the second is the application of combining a kang with other technologies [58]. Zhuang et al. [59] used the heat flow method to simulate the heat transfer process of flue gas entering the body, focusing on analyzing the flue gas flow process to avoid energy waste caused by flue gas backflow. The use of a new type of flue can effectively avoid the backflow of flue gas. Zhao et al. [60] put forward a new system for solar heating kang, which has the advantage of collecting heat throughout the day. The indoor temperature can be increased by approximately 13 °C, and the surface temperature of a kang can reach 26 °C, which has a good heating effect. Wei et al. [61] combined a kang with a solar water collector, established a mathematical model, and determined that the water inlet temperature of a kang can reach 35–55 °C.

Passive heating in the above three aspects has been proven to raise the indoor temperature and decrease the demand for energy, but there are still some problems. In the study of improving the phase-change material of enclosure structures, most buildings in rural areas are restricted by factors such as the building structure and cost, and it is difficult to improve the phase-change material of a maintenance structure. At the same time, phase-change materials only improve the process of heat transfer, not from the heat source; for all the research, there is still a lack of energy saving. In addition to considering the construction cost when using solar energy, it is also necessary to determine whether the weather influences the solar radiation on the indoor temperature. In terms of biomass combustion, a large amount of carbon will be produced during the combustion process, which will lead to environmental pollution. Therefore, proposing a firewall system with an additional sunlight room system, which can effectively raise the indoor thermal comfort in winter, can achieve energy saving.

### 1.3. Scientific Originalities

In previous studies, a new passive heating design strategy for rural residential buildings has been proposed, combining the characteristics of cooking and heating methods, and studying the basic working principles of firewall heating and energy saving [62]. Based on this principle, this research added new heating equipment, combined the cooking firewall system with solar energy technology, and comprehensively analyzed the heating time, effect, and energy saving.

First of all, in previous research, the time for cooking and heating and the heat transfer of the wall have been analyzed and discussed. The main purpose was to extend the thermal time and improve the heating effect. The specific method was based on the firewall cooking and heating system, and a glass sun room was built in conjunction with the kitchen on the sunny side of the building. Its function is to collect the heat generated from cooking and prevent the accelerated loss of hot air in the room. The principle is that the hot air is stored in the glass sunlight room through the heat transfer of the wall, while preventing direct contact between the hot air and the outdoor cold air, causing the indoor temperature to drop rapidly, similarly to the design principle of a thermos cup.

Second, the use of the greenhouse effect makes the internal temperature of the glass sunlight room increase when receiving solar radiation and keeps the temperature constant. The principle is that the airtight space between sunlight forms a heat preservation effect due to the lack of heat exchange with the outside, so as to achieve the effect of heating and heat preservation. At the same time, it can also make up for the discontinuity of cooking and heating in time. On the basis of whether the wall has solar radiation as a separate reference, systematic analysis and research on whether the new system takes this factor into consideration was carried out, which verifies the importance of whether the wall has solar radiation for winter heating.

Finally, compared to a separate cooking and heating system, the new system also has functions such as building wall insulation and the greenhouse effect of the glass sun room. In essence, this study used completely passive heating technology. The results showed a greater reduction in the heat load of the building compared to previous research, as well as an improvement in the heating effect to a new level. This makes up for the blank passive comprehensive energy-saving design in rural residence.

### 1.4. Aim of the Study

For increasing the indoor temperature of residences and reducing the heat load of the building, this research proposes a passive heating system between a firewall and an additional sunlight room. Based on the model, the new system was added for comparison with traditional residential buildings in various aspects. The specific research objectives were as follows:By comparing the heating conditions of traditional residential buildings with or without the new passive system, the heat load reduction when adopting the new system in winter was estimated.In the absence of solar radiation, the heating effects of the ordinary heating system and the new passive system were analyzed. The purpose was to calculate the heating rate, the time required, and the decrease in the annual heat load of the building.Considering the presence or absence of solar radiation, the new system was compared to the ordinary heating of residential buildings to simulate and calculate the change in indoor temperature, heating efficiency, heating time, and the value of the annual building heat load.Through software simulation computation, the appropriate use time of the whole year after adopting the new system was estimated.According to the standard minimum value of indoor thermal comfort temperature, under basic heating, the total time to meet the minimum value of the comfort standard throughout the year was calculated. Compared to the use of the new system with or without solar radiation, the total time to meet the lowest value of the comfort standard throughout the year was estimated.

## 2. Methodology

### 2.1. Basic Research

#### 2.1.1. Overview of the Current Situation

The research site was Hong Village, Houliu Town, Shiquan County, Ankang City, Shaanxi Province, covering an area of approximately 0.01625 km^2^ (Figure 1a). The development of the village along the Han River is ribbon-shaped, and the terrain is mostly mountainous. This area is hot and humid in summer with more rainstorms, and cold and humid in winter with less rain. According to the field survey, the lowest temperature is –2 °C, and the highest wind velocity is 5 m/s (Figure 1b,c). Due to the constraints posed by the river valleys and mountains, the distribution of villages is mainly in clusters, and the layout of residential buildings is relatively close. The residential buildings in the Ming and Qing dynasties were mostly of the one- or two-floor, sloped roof type. Among them, the one-story residential houses accounted for 88.2% of the total, with a floor height of 4–4.5 m; meanwhile, the two-story residential houses had a floor height of 5.5–6.0 m, and the building density was 23.15% (Figure 1d). Due to the influence of the building form and structure, the indoor solar radiation received by the residential houses are relatively small, and the influence of the climate causes the residential houses to feel damp and cold in winter.

#### 2.1.2. Research Method

The research methods were as follows:

Field research: This refers to the use of questionnaires, photographs, measurements, and other research methods to analyze the research objects. The thermal environment of the traditional residential buildings in southern Shaanxi and the thermal properties of the materials were studied in depth. Through the use of analyses, comparisons, inductions, and other research methods, the representative traditional residential houses in southern Shaanxi were taken as the research objects.

Data collection: The data of the field investigation were analyzed and summarized. Data on indoor and outdoor temperatures, cookware sizes, smoke temperature, room sizes and areas, and door and window positions and sizes were collected and classified.

Induction and summary: The survey data were classified and summarized in light of certain requirements. Through analysis and comparison, the basic characteristics of the residence in southern Shaanxi were summarized. In addition, the indoor and outdoor temperatures, the size of the cooker, the temperature of the flue gas, the size and area of the rooms, the position and size of the doors and windows, and other data were also summarized in accordance with certain requirements.

#### 2.1.3. Research Content

The research objects were the traditional dwellings in southern Shaanxi. The buildings of the folk houses are primarily one-story, although some parts of the houses are two-story. The overall height of the buildings is 4.8 m; the height of the first floor is 3.3 m and the height of the local second floor is 1.5 m. The walls of the building are made of blue bricks; the roof has double slopes of blue tiles and the eaves are longer, which is conducive to blocking the sun and organizing rainwater drainage. Local dwellings are mostly self-built, and the construction method is based on years of construction experience. Restricted by economic conditions and building materials, less consideration is given to the passive heating of residential buildings in winter, resulting in lower indoor temperatures in winter (Figure 2).

For effectively increasing the indoor temperature in winter, residents use different heating equipment, such as braziers and fire tables. These heating devices can only increase the temperature of a part of the room and are mostly used in the daytime, and have disadvantages such as a poor heating effect and inflexible use. This paper presents a firewall heating system with an additional sunlight room, which consists of two parts. The first part is the firewall; its working principle is to use the heat storage capacity of the wall to heat said wall, with the residual temperature generated by cooking raising the indoor temperature (Figure 3a). The second part is the additional sunlight room. Since the heating time of the firewall mainly depends on the duration and frequency of cooking, in order to solve this problem, an additional sunlight room was designed on the basis of the firewall (Figure 3b). The additional sunlight room refers to a closed space made of transparent materials such as glass. When solar radiation is irradiated in this transparent and enclosed space, the ability of the wall and the ground to absorb and release heat is used to heat the indoor temperature in the form of thermal radiation and thermal convection, thereby prolonging the indoor heating time (Figure 3c). This research mainly used the advantages of the firewall system and the sun room system to make up for the shortcomings of a single system, so as to improve heating efficiency and energy saving.

The key technology to be solved in this study was how to use firewalls and additional sunlight for passive heating to raise the indoor temperature. By analyzing the local winter sunshine, it is known that when designing a firewall system with an additional sun room, the sun room should be combined with the kitchen, so that the sun room can collect the waste heat generated by cooking (Figure 3d–f). Second, the sun room should be designed on the side facing the sun, so that the sun room can fully collect and receive solar radiation. At the junction of the pipe and the furnace body, two valves are provided. In winter, valve 1 is opened and valve 2 is closed, letting the flue gas pass through flue 1 to be exhausted outdoors and to heat flue 1 (Figure 3g). In summer, valve 2 is opened and valve 1 is closed, allowing the flue gas to pass through flue 2 to the outside (Figure 3h). The additional sunlight room system needs to be equipped with sliding glass to prevent the indoor temperature from becoming too high in summer.

### 2.2. Cooking Heating and Additional Sunlight Room

#### 2.2.1. Design of Cooking Heating and Additional Sunlight Room

Based on comprehensive research and comparison, residential buildings with four bays and one depth were selected as the object of this study. According to the established model, the spatial distribution from west to east is the bedroom, the living room, the bedroom, and the kitchen. The reformed firewall plan is as follows: The firewall is set between the bedroom and the kitchen on the east side (Figure 3a). The specific structure is as follows: A circular hole with a diameter of 100 mm is set inside the hearth, and an aluminum tube with a thickness of 3 mm is inserted into the hole and placed inside the wall in an S shape, with an overall length of 11.37 m. After renovation, the plan of the additional sunlight room is as follows: A 1.5 m × 3.0 m × 19.34 m confined space is built on the sunny side of the dwelling with the kitchen, which is the additional sunlight room. As shown in the purple area, the outer protective structure is composed of glass curtain walls (Figure 3b). The glass adopts a single-layer glass of 6 mm, and the visible light transmittance is 0.77. The combined design of cooking heat and sunlight can not only make up for the lack of cooking-based heating time but can also meet the overall needs of heating the indoor temperature.

#### 2.2.2. Working Principle of Cooking-Based Heating and an Additional Sunlight Room

The working principle of the firewall is that the hot gas generated by cooking enters the aluminum tube heating wall, and the tube wall heats the wall after reaching a constant temperature. Using the heat storage capacity of the wall, the indoor temperature is heated to enhance the thermal comfort of the human body. The design of the firewall is similar to the cooking waste heat recovery equipment. For better analyzing the heat transfer process of the firewall, the heat transfer process includes three steps: First, the material burned during cooking produces high-temperature gas, and the hot gas enters the aluminum tube to heat the tube wall. The entire heat transfer process is dominated by heat convection, which belongs to the coupled heat transfer between fluid and solid states. In the process of software simulation, hot air enters the aluminum tube and forms a turbulent flow. When the speed of the hot gas reaches a relatively constant value, the hot gas heats the tube, and the heat transfer mainly takes the form of heat convection. When the tube is heated to a constant temperature, the wall is heated in the form of heat conduction and thermal radiation. This process is a solid–solid coupling heat transfer. In addition, the heated wall heats the interior of the wall in the form of heat radiation and convection to raise the temperature. Finally, when the cooking is over, no more smoke is produced, the temperature of the firewall begins to drop, and the heat transfer process also ends.

The working principle of the additional sun room is that the sun heats the air in the sun room through the glass and irradiates the wall and the ground to absorb and store part of the heat energy. The other part of the sunlight can be directly irradiated into the room through the windows, thereby increasing the indoor temperature. From the perspective of the indoor heating principle, the mechanism is the same as the principle of the heat collection and storage wall type, which is a combination of the direct revenue type and the heat collection and storage type. In order to understand the whole heat transfer process, it is split into three steps: First, sunlight heats the temperature in the room through the glass. This heat transfer process is mainly in the form of heat transfer by heat radiation and convection. The time sequence in the heat transfer process is transient heat transfer–steady heat transfer–transient heat transfer. When the fluid is heated, it follows Newton’s law of cooling, and its calculation formula is
(1)$${q=h(T}_{w}-T_{f})$$
where formula h is the surface heat transfer coefficient, T_w_ and T_f_ are the wall temperature and fluid temperature, respectively, and q is heat flux.

When the temperature reaches a certain level, the wall begins to absorb and store heat. The heat transfer in this process is mainly heat conduction and heat convection. In heat conduction, Bouriet’s law is followed and its calculation formula is
(2)$$q=\frac{Q}{A}=-\lambda (\frac{\mathrm{dt}}{\mathrm{dx}})$$
where formula Q is the heat conduction rate, Q is the heat flux, and λ is the proportionality coefficient. When the temperature t increases along the x direction, dt/dx > 0 and q < 0, indicating that the heat is transferred in the direction where X decreases. On the contrary, dt/dx < 0 and q > 0 explain that heat is transferred along the increasing direction of the x-axis at this time.

When the indoor temperature is lower than the temperature of the room with additional sunlight, the high temperature exterior of the wall conducts heat to the low-temperature indoor wall. The temperature between the walls is heated in the form of thermal convection and thermal radiation, thereby increasing the overall temperature. In heat radiation, it follows the fourth power law and its calculation formula is
(3)$$\Phi ={\varepsilon A\sigma T}^{4}={\varepsilon C}_{0}(\frac{T}{100})$$
where formula Φ is the radiation energy, T is the thermodynamic temperature, A is the radiation surface area, ε is the emissibility of the object, σ is the constant 5.67 × 10^−8^, and $C_{0}$ is the constant 5.67.

#### 2.2.3. Ordinary Heating

The ordinary heating of residential buildings mainly relies on the heat radiation of the sun through the heat transfer of the walls to raise the indoor temperature. The heat transfer process is as follows: The wall absorbs heat and solar radiation through thermal convection and thermal radiation as the main heat transfer methods. Second, the wall self-generates heat conduction, and the high-temperature outer surface transfers heat to the low-temperature inner surface. The inner surface of the wall radiates heat between the walls, thus heating the indoor temperature. In short, ordinary heating is a synthesis of heat convection, conduction, and radiation. In the process of basic heating, the indoor temperature changes significantly, and the indoor temperature differs from the outdoor temperature. Moreover, the change in indoor temperature should also be considered when there is no wall solar radiation (WSR). The indoor thermal disturbance is the main source of indoor temperature change. The heat disturbance mainly includes personnel activities, household appliances, room size, window size, and other such factors. For making the simulation results more compatible with the actual situation and therefore usable, does not have the ordinary heating of WSR, and the indoor temperature changes in a small range, which is different from the outdoor temperature.

#### 2.2.4. Heating between the Firewall and the Additional Sunlight Room

In order to effectively improve the limitation of cooking and heating time, a passive firewall and additional sunlight heating are proposed. Not only can the heating time be extended, but it can also make up for the insufficient heating effect of the additional sunlight on cloudy days. When the sun shines on the additional sunlight room, the temperature starts to rise slowly, and cooking has stopped by this point. When the temperature in the additional sunlight room reaches a relatively constant value, the indoor temperature is heated through the heat absorption and heat release effect of the wall. The wall, through the form of heat radiation and convection, raises the temperature of the room. Compared to single-firewall heating, the passive firewall heating and additional sunlight heating effect is better; after the balance of the temperature has been significantly increased, the heating time is extended. In addition to considering the presence of WSR, the absence of WSR is also considered. In the absence of WSR, the temperature between the wall and the additional sunlight room rises slowly. When the temperature in the sunlight room reaches a relatively constant value, the indoor temperature starts to rise slowly. In the absence of solar radiation, compared to a separate firewall system, the firewall and additional sunlight room combination is more effective in raising the indoor temperature. In the absence of solar radiation, a firewall combined with additional sunlight is more effective at raising the indoor temperature than a firewall system alone.

### 2.3. Simulation Steps

The simulation software included ANSYS (American ANSYS company, Pittsburgh, America), Open Studio (National renewable energy laboratory, Golden, CO, USA), Ladybug (National renewable energy laboratory, Golden, CO, USA), Energyplus (National renewable energy laboratory, Golden, CO, USA), and MATLAB (MathWorks, Natick, MA, USA). The boundary condition was the presence or absence of solar radiation. ANSYS was mainly used to calculate the heating time and the indoor temperature after the heating balance. Open Studio, Ladybug, and Energyplus mainly calculated the reduction in the heat load of residential buildings throughout the year by using different heating methods. The annual weather data came from the Energyplus weather data package. The annual radiation range was 150–280 MJ·M^2^ (Figure 4a), the temperature range was −2 to −34 °C, and the relative humidity range was 22–95% (Figure 4b). ANSYS was mainly used for turbulence calculation, steady-state and transient thermal calculation, thermal convection, and thermal radiation calculation. The turbulent calculation primarily simulated the hot gas entering the tube wall and calculated the temperature of the tube and the required time. The steady-state and transient thermal calculation mainly computed the temperature and time of the tube wall heating the wall, as well as the heat transfer time and temperature of the wall between additional sunlight. Heat convection and radiation mainly computed the temperature in the firewall heating room, the time and temperature of the additional sunlight heated by solar radiation, and the situations that affect the indoor temperature. The heat load of buildings was calculated with different heating methods. Open Studio, Ladybug, Energyplus and MATLAB were used to simulate the calculations. The steps were to establish a simulation in Open Studio, to put the built model into Ladybug software, and to assign material attributes. The model was carried into Energyplus for simulation analysis, and the heat load results were imported into MATLAB for editing.

This simulation mainly includes three parts. Part one was to build a model based on the data measured on the spot, and to set up a firewall model, which mainly included the stove, aluminum tube, and wall. The content of the simulation calculation was mainly the temperature and time required for the aluminum tube, the firewall, and the indoor temperature and time after equilibrium. In the second part, based on the original firewall model, the additional sunlight interval model was established, including the size and area of the sunlight interval. The content of the simulation was mainly the temperature between the sunlight, the heat transfer of the wall, and the indoor temperature and time after the balance. The third part mainly involved simulating and calculating the annual heat load of the new system and the appropriate time. Table 1 includes the basic information of the dwellings in southern Shaanxi, such as the climate conditions, schedule, number of occupants, size of houses, and construction techniques, such as the walls and roofs. Table 2 is the related properties of the building materials.

#### 2.3.1. Establishing Grids

The model was established according to the measured data and imported into ANSYS. According to the method provided by Arturs et al., the grid was set and divided, and finite element simulation analysis technology was used to analyze the indoor temperature [63]. The k-epsilon format was selected for the model, which is conducive to a more stable model analysis. For ensuring the simulation results were close to the actual situation, it was necessary to set the convergence value. Herein, the convergence value of temperature and energy was 6–10, and the convergence value of the other values was 4–10. The grid setting in ANSYS software is very important, which determines the accuracy of the numerical simulations. Arturs et al. [63] used 1.54, 1.92, 0.55, 0.98, 2.09, and 0.34 million hexagonal sections to divide the grid. In this research, mapped meshing and local refinement were selected. In the process of mapping the grid division, it was divided into 5.324 million cells. In order to reduce the simulation time, the grid size can be controlled by setting the grid size value and the grid growth rate. When setting the grid between the wall and the roof, the grid should be evenly arranged on the roof and the wall. This is conducive to the simulation, with results being closer to the actual situation. As the walls are made of four layers of material, the thickness was 251 mm, and they were kept at 4.5% between adjacent grids.

#### 2.3.2. Interface Conditions

The energy equation was used and k-epsilon and RNG (National Renewable Energy Laboratory, Golden, CO, USA), were used in the turbulence model. The standard in near-wall treatment and buoyancy effects in options was selected. In the thermal radiation model, automatic calculation of the appropriate amount of the sun’s irradiation direction was chosen, and the local longitude, latitude, and date were entered. When setting the material, the relevant thermal properties of the walls, glass, air, and other materials were entered.

#### 2.3.3. Boundary Conditions

After setting the abovementioned parameters, the boundary conditions, such as the walls, glass, floors, floors, and fluids, were set up. Then, detailed parameters in the five boundary conditions were set, including the initial temperature, material, and radiation absorption rate. Finally, the solution was conducted.

## 3. Results and Discussion

In order to verify the effectiveness of the firewall and the additional sunlight room system, ordinary heating was used as a benchmark for comparison to the new system. Ordinary heating refers to keeping the body of the residential house unchanged, and only relying on solar energy to heat the interior. The new system is the firewall system with an additional sunlight room mentioned in a previous article [62], which refers to the renovation of residential buildings, adding firewalls and additional sunlight rooms. Heating the interior using the heat generated by the firewall from cooking while relying on absorption of additional sunlight and storing and converting solar energy, so as to heat the interior more effectively.

In order to make the test and experiment results accurate, the room and windows were kept closed during the test and experiment, forming a relatively closed space indoors. At the same time, during the simulation process, the room was set up as an independent closed fluid space, thereby reducing the influence of outside temperature on the room.

### 3.1. Comparison of Ordinary Heating with or without WSR

#### 3.1.1. Ordinary Heating with WSR

To confirm the effectiveness of the firewall system with an additional sunlight room, a comparison was made to ordinary heating as the benchmark. For verifying the practicability of the results of the software simulation, the indoor temperature was measured for three consecutive days. The test took place from 25 to 28 December 2020. The test interval was measurement every 1 h. The testing equipment included four thermometers and hygrometers (type: PM6508) and four rangefinders (type: D508). The indoor measuring points were mainly selected in the middle of the room, and the outdoor measuring points were mainly selected at 2 m away from the building (Figure 5a). During the test, the sensor of the instrument was perpendicular to 1.5 m on the ground, and the data were read after standing for 1 min. The test results were as follows: From 2:00 to 4:00 p.m. in the afternoon, the highest outdoor temperature was 5.0 °C and the average temperature was 3.5 °C (Figure 5b). From 9:00 to 11:00 a.m. in the morning, the average indoor temperature rose from 5.0 to 5.4 °C. At 2:00 p.m., the average indoor temperature rose to 6.4 °C, and by 5:00 p.m., the indoor temperature began to drop. At 7:00 p.m., the indoor temperature was 4.4 °C (Figure 5c). In order to make the simulation more realistic, through the comparison of multiple sets of data, the day that was least affected by solar radiation in the winter was selected and the initial indoor temperature was 5 °C (Figure 5d). In order to compare the situation to solar radiation, a simulation time during the day was selected, and the simulation time was mainly from 9:00 a.m. to 7:00 p.m. The main reason is that the wall is affected by the solar thermal radiation, which causes the indoor temperature to rise. From 9:00 to 11:00 a.m., under the influence of solar radiation, the indoor temperature began to rise slowly, from an average temperature of 5 to 5.5 °C (Figure 6a,b). At 3:00 p.m., the solar radiation altitude angle changed, and the average indoor temperature rose to 6.5 °C, which is 1.9 °C different from the outdoor temperature (Figure 6c,d). From 5:00 p.m., the indoor temperature began to slowly drop, and the average temperature was 6 °C (Figure 6e). At 7:00 p.m., the average indoor temperature dropped to 4.5 °C (Figure 6f). Therefore, ordinary heating mainly relies on solar radiation to improve the indoor temperature through the irradiation of walls, glass, and other structures. By comparing the software simulation results with the actual measurement results, it can be seen that the temperature difference between the two is 0.1 °C (Figure 7). Within the allowable range of error, it was proven that the results of the software simulation were close to the actual situation.

#### 3.1.2. Ordinary Heating without WSR

In addition to the presence of WSR, the absence of WSR should also be considered. The simulation time, method, and initial indoor temperature are consistent with the previous situation with solar radiation. According to the software simulation, the indoor temperature changed slowly. From 9:00 a.m. to 12:00 p.m. (noon), the indoor and the outdoor temperatures basically remained unchanged, and the average indoor temperature dropped from the initial 5 °C to 4.5 °C (Figure 8a,b). At 3:30 p.m., the indoor temperature began to rise slowly, and the average indoor temperature was 5.5 °C (Figure 8c). With the time increasing from 4:00 to 7:00 p.m. in the evening, the indoor temperature began to drop. The average indoor temperature reached a minimum of 4.6 °C at 7:00 p.m., a difference of 0.1 °C compared to outdoors (Figure 8d). The results showed that if ordinary heating has no radiation, the indoor and outdoor temperatures are similar.

#### 3.1.3. Comprehensive Analysis

For ordinary heating, the heating time is 6.5 h, the temperature after equilibrium is 6.5 °C, which is 1.9 °C different to the outdoor temperature, and the heating efficiency is 0.23 °C/h. In the case of ordinary heating without WSR, the heating time is 7.3 h, the temperature after equilibrium is 5.5 °C, which is 0.5 °C different to the outdoor temperature, and the heating efficiency is 0.06 °C/h. If ordinary heating has no radiation, the indoor and outdoor temperatures are similar. Ordinary heating is mainly affected by solar radiation, so there are requirements for weather conditions. This heating method has no obvious effect on increasing the indoor temperature.

### 3.2. Comparison of the New System with or without WSR

#### 3.2.1. New System with WSR

In order for a comparison with ordinary heating, the indoor and outdoor temperature distributions were basically the same when the firewall was simulated and the sun room was added. The heating of the firewall is inseparable from the energy released when cooking. After the cooking has started, the firewall starts to work. The combustion of cooking is mainly divided into four parts: firewood burning, firewood burning, stop adding firewood, and firewood burning out. According to the actual measurement, when the firewood is ignited, the flue gas enters the aluminum pipe at a speed of 2.3 m/s. The software simulation shows that the average velocity of gas in the aluminum pipe is 2.26 m/s and the average pressure is 1.56 Pa (Figure 9a,b). The temperature of the gas depends on the energy of cooking and burning biomass. Through multiple measurements, the average temperature of the gas was shown to be 60 °C. Taking 60 °C as the initial temperature of the gas, software simulations can be conducted. At 25 s, the temperature at the inlet of the pipe was 60 °C and the outlet was 0 °C (Figure 10a). As the time increased, at 65 s, the mean temperature of the pipe was 28.32 °C (Figure 10b). At 95 s, the mean temperature of the pipe was 42.62 °C (Figure 10c). At 125 s, the temperature of the pipe reached equilibrium, and the final temperature was 60.22 °C (Figure 10d).

When the aluminum tube reached a constant temperature, the firewall is heated by heat conduction. At 4 min and 43 s, the average temperature of the firewall was 4.38 °C (Figure 11a). At 7 min and 45 s, the average temperature of the firewall was 9.43 °C (Figure 11b). With the increase in heating time, the average temperature of the firewall was 14.49 °C at 11 min and 30 s (Figure 11c). At 16 min and 10 s, the average temperature of the firewall gradually increased to 19.55 °C (Figure 11d). At 20 min and 10 s, the average temperature of the firewall was 24.60 °C (Figure 11e). At 24 min and 20 s, the temperature of the firewall reached a stable value of 29.66 °C, and the temperature of the additional sunlight room remained unchanged (Figure 11f).

Through the investigation of the duration of cooking in winter, a single cooking event took 60 min, where the initial firewall heating time was 26 min and 25 s, and the remaining time was 34 min and 35 s. In order for a comparison to ordinary heating, the simulation started at 9:00 a.m., and the initial temperature was also 5 °C. Under the condition of solar radiation, the adjacent firewall room began to rise slowly at 9:10 a.m., from the initial temperature of 5 °C to 7.24 °C, and the temperature in the room with additional sunlight remained basically unchanged (Figure 12a). At 9:20 a.m., the temperature of the adjacent firewall room rose to 9.16 °C, and the temperature in other rooms began to rise slowly (Figure 12b). At 9:30 a.m., the temperature of the adjoining firewall room rose to 11.07 °C, and the temperature in the other rooms was 8.2 °C (Figure 12c). At 9:34 a.m., the temperature of the adjacent firewall room rose to 13.56 °C, while the temperature in other rooms was 9.12 °C, and the temperature in the additional sunlight room began to slowly rise to 6.23 °C (Figure 12d). When the cooking system stopped, the room temperature remained unchanged within 30 min; meanwhile, the temperature of the additional sunlight room started to rise slowly. At 10:10 a.m., the temperature in the additional sunlight room was 7.55 °C, and the temperature in the other rooms dropped to 6.14 °C (Figure 12e). At 10:30 a.m., the temperature in the additional sunlight room was 8.16 °C, and the temperature in the other rooms began to slowly rise to 6.32 °C (Figure 12f). At 10:50 a.m., the temperature in the additional sunlight room was 10.39 °C, and the temperature in the other rooms began to slowly rise to 6.42 °C (Figure 12g). At 11:20 a.m., the temperature in the room with additional sunlight reached a stable value of 13.99 °C, and the temperature in the other rooms began to rise slowly to 6.53 °C (Figure 12h). At this time, the heat transfer of additional sunlight through the wall heated the indoor temperature. At 11:36 a.m., the average indoor temperature was 7.55 °C (Figure 12i). At 2:16 p.m., the indoor temperature rose to the highest value of 9.16 °C (Figure 12j). With the increasing time, the temperature between the additional rays began to drop. At 6:00 p.m., the temperature in the additional sunlight room was 10.55 °C, and the temperature indoor was 8.73 °C (Figure 12k). At 6:24 p.m., the temperature in the additional sunlight room was 9.43 °C, and the indoor temperature changed little (Figure 12l). At 6:45 p.m., the temperature in the additional sunlight room was 7.88 °C, and the indoor temperature dropped to 9.73 °C (Figure 12m). At 7:00 p.m., the temperature in the additional sunlight room was 6.93 °C, and the indoor temperature dropped to 6.53 °C (Figure 12n). At this time, there was a difference of 2.53 °C from the outdoor temperature.

#### 3.2.2. New System without WSR

The simulation of the new system also started at 9:00 a.m., and the initial indoor temperature was also 5 °C. At 9:10 a.m., the temperature of the adjacent firewall room began to rise slowly, from the initial temperature of 5 °C to 5.52 °C (Figure 13a). Meanwhile, the temperature of the additional sunlight room remained basically unchanged. At 9:20 a.m., the temperature of the adjoining firewall room rose to 6.03 °C, and the temperature in other rooms began to rise slowly (Figure 13b). At 9:30 a.m., the temperature of the adjacent firewall room rose to 6.53 °C, and the temperature in the other rooms was 5.84 °C (Figure 13c). At 9:34 a.m., the temperature of the adjacent firewall room rose to 8.79 °C, the temperature in other rooms was 7.41 °C, and the temperature in the room with additional sunlight remained basically unchanged (Figure 13d). When the cooking system stopped, the room temperature within remains unchanged15 min, and the temperature of the additional sunlight room started to rise slowly at this time. At 9:45 a.m., the temperature in the additional sunlight room was 5.44 °C, and the temperature in the other rooms dropped to 6.64 °C (Figure 13e). At 9:55 a.m., the temperature of the additional sunlight room was 5.45 °C, and that of other rooms is 6.32 °C (Figure 13f). At 10:10 a.m., the temperature in the additional sunlight room remained basically unchanged, and the temperature in the other rooms was 5.57 °C (Figure 13g). At 10:20 a.m., the temperature in the additional sunlight room reached a stable value of 5.45 °C, and the temperature in the other rooms was 5.43 °C (Figure 13h). This is mainly because the temperature in the additional sunlight room mainly depends on solar radiation. When there is no solar radiation, the indoor temperature range is small. At 6:00 p.m., the temperature in the additional sunlight room began to drop to 5.11 °C, and the indoor temperature was 5.23 °C (Figure 13i). At 6:24 p.m., the temperature in the additional sunlight room was 4.98 °C, and the indoor temperature was 5.11 °C (Figure 13j). At 6:45 p.m., the temperature in the additional sunlight room was 4.73 °C, and the indoor temperature dropped to 4.86 °C (Figure 13k). At 7:00 p.m., the temperature of the additional sunlight room and indoor remained basically unchanged (Figure 13l). At this moment, there was a difference of 0.86 °C to the outdoor temperature.

#### 3.2.3. Comprehensive Analysis

Due to the heating system between cooking and additional sunlight, the heating time is intermittent. First, the firewall system was 26 min and 25 s in the early stage, and the remaining time was 34 min and 35 s. At this time, the temperature of the additional sunlight room remained basically unchanged. The indoor temperature basically stayed the same within 30 min after the cessation of cooking activities. The temperature of the adjacent firewall room was 13.56 °C, and the other rooms were kept at 9.12 °C. The average indoor temperature was 11.34 °C, and the temperature increased by 6.34 °C. With the increase in temperature in the additional sunlight room, after 4 h and 16 s, the indoor temperature was 9.16 °C, with a 4.76 °C difference to the outdoor temperature, and the temperature increased by 4.16 °C. In winter, relying on ordinary heating alone is not enough to meet the thermal comfort requirements of the human body, mainly because the heating time is long and the temperature change is not obvious.

In the absence of WSR, the indoor temperature remained basically unchanged within 15 min after the cessation of cooking activities. The temperature on both sides of the firewall was 8.79 °C, the temperature in other rooms was 7.41 °C, and the mean indoor temperature was 7.95 °C, increasing by 2.55 °C. With the increase in temperature in the additional sunlight room, the indoor temperature was 5.43 °C after 1 h and 20 min, which was 1.43 °C different to that outdoors—an increase of 0.43 °C. Ordinary heating systems mainly rely on WSR. Yet, when cooking and additional sunroom systems are used, the heating time is greatly increased. Without considering the WSR, the mean heating efficiency between cooking and additional sunlight was 2.87 °C/h. Considering WSR, the mean heating efficiency between the cooking heat and additional sunlight was 4.09 °C/h.

### 3.3. Heat Load Reduced by the Firewall–Sunlight System

The annual heat load of the firewall system with the additional sunlight room was simulated and calculated by Energyplus software. When there was WSR, the annual heat load reduction of the building was 1436.731 kW· h (Figure 14a), a decrease rate of 20.21%. When there was no WSR, the annual heat load reduction of the building was 735.919 kW· h, a decrease rate of 8.56%. The thermal load difference between the two was 840.812 kW· h (Figure 14b).

### 3.4. Suitable Year-Round Time for the Use of the Firewall–Sunlight System

The presence or absence of WSR between the firewall and the additional sunlight room affects the temperature of the room after heating, the extension of heating time, the efficiency of heating, and the reduced heat load. For this system, the time required for heating using the firewall system with an additional sunlight room, the change in indoor temperature, and the decreased heat load value were estimated for the system in the presence or absence of solar radiation throughout the year.

The suitable times of the year for using the system are mainly distributed in January, February, March, November, and December. In the presence of WSR, the annual firewall and additional sunlight room system were used for 1155 h, accounting for 13.11% of the total time of the year (Figure 15a). In the absence of WSR, the firewall system with an additional sunlight room was used throughout the year for 882 h, accounting for 10.01% of the total time of the year (Figure 15b).

### 3.5. Adopting the Standard Time of Thermal Comfort of the Firewall–Sunlight System

Software simulation was used to calculate the thermal comfort range of residential houses. The indoor thermal comfort was significantly improved under the action of the new system. In this study, the three cases of ordinary heating and the new system with or without WSR were compared and analyzed, and the time for each room to be below 18 °C was calculated. Taking 18 °C as the standard value of the building’s winter thermal comfort, the proportion of time when the temperature was lower than 18 °C throughout the year was calculated. In the case of ordinary heating, the annual temperature below 18 °C was 4532 h, accounting for 51.45% of the total annual time (Figure 16a). In the presence of WSR, when the firewall system with an additional sunlight room was used, the time when the temperature was lower than 18 °C was 3733 h, which is a decrease of 799 h, meaning a reduction rate of 17.63%, accounting for 38.34% of the total time of the year (Figure 16b). In the absence of WSR, the annual time below 18 °C was 4006 h, which is a decrease of 526 h, meaning a reduction rate of 11.60%, accounting for 41.43% of the total annual time (Figure 16c). In addition, the time each room was below 18 °C was also calculated. Due to the different location of the room, the indoor time below 18 °C was also different. The indoor time below 18 °C in rooms 1 and 2 was significantly reduced by adopting the new system compared to the ordinary heating system. The time below 18 °C in rooms 3 and 4 was also reduced by adopting the new system compared to the ordinary heating system, and the reduction was lower than that in rooms 1 and 2 (Figure 16d).

### 3.6. Optimization of the New System Heating

#### 3.6.1. Window-to-Wall Ratio

Obrecht et al. [64] showed that the ratio of wall to window is one of the important parameters affecting indoor temperature. Therefore, the influence of different window-to-wall ratios on the thermal comfort time of the new system is discussed. First, to determine the scope of the window-to-wall ratio: According to Chi et al. [65], the range value was determined to be 0.1–0.9, and the division standard is divided in increments of 0.1. The window-to-wall ratio of the model was also determined to be 0.1–0.9, and the model is divided into 9 groups of window-to-wall ratio models in increments of 0.1. Secondly, Chi et al. [65] verified the accuracy of the software simulation by comparing the measured data with the simulated data of the software Energyplus. Therefore, EnergyPlus software is also used in this simulation. Finally, through software simulation, the influence of different window-to-wall ratios under the new system on the annual thermal comfort time was analyzed. The research results show that when the window-to-wall ratio is 0.4, the new system has the least thermal comfort time below 18 °C throughout the year (Figure 17).

#### 3.6.2. Wall Materials

The heat transfer of the fire wall in the new system depends on the insulation and heat storage performance of the material. Therefore, the wall material should be optimized. In order to improve the compressive strength of the concrete, according to Farzampour et al. [66], some manufacturers add rubber and additives of different materials. Thus, the durability and compression resistance of the concrete are prolonged. Although these materials are low in cost, it can pollute the environment. Farzampour et al. [67] found that blast furnace slag and fly ash can improve its resistance, but the cost is high and the use of materials is limited. Mansouri et al. [68], by replacing part of the cement with a mixture of nano-silica and metakaolin at a ratio of 1% and 10%, respectively, improved the compressive strength of concrete by 15%. Navid et al. [69] found that the use of a combination of nano-silica and metakaolin superseded the effect of rubber on concrete. The results show that the workability of concrete is improved by the combination of the two materials. In addition, the cost of these two materials is relatively low. During installation, in order to improve the thermal insulation of the firewall, common thermal insulation materials such as polystyrene board and foam cement board are selected. Dong et al. [70] showed that when installing polystyrene boards, the first step is to treat the foundation to remove the wall concrete residue and release agent. Then it is to control the line and hang the reference line and configure the adhesive polymer mortar. The bonded area between the polystyrene board and the wall reaches 40% with an adhesive and is installed and polished with anchor pieces. A 2–3 mm mortar plaster size was selected and pressed with a glass fiber mesh cloth. Finally, concrete plaster with nano-silica and metakaolin was selected. The installation process is relatively simple, and it is necessary to pay attention to the treatment of the base layer to remove the oil, dust, and other attached substances on the wall. Jin et al. [71] analyzed the installation of foamed cement boards, and firstly dealt with the base wall and the control line and the datum line: Install the base bracket and 20 mm cement mortar to treat the wall surface, and fix the foam cement board with adhesive and anchor bolts. The alkali-resistant fiber mesh cloth was pressed with 3–6 mm plaster mortar, and finally the concrete plaster with nano-silica and metakaolin was selected. The installation process is relatively simple, but attention should be paid to the impact of rainy days on the foam cement board.

## 4. Conclusions and Future Prospects

### 4.1. Conclusions

This research focused on the passive heating of residences in southern Shaanxi in winter, proposing a firewall–sunlight heating system combining the cooking method and solar heating in southern Shaanxi. The firewall system uses the hot air generated by cooking to heat the wall and raises the indoor temperature in the form of heat radiation and convection. The sunlight room system establishes an additional sunlight room on the sunny side of the building, using the principle of the greenhouse effect to improve the heating efficiency. The new system combines the advantages of the firewall system and the sun room system to further improve the indoor temperature. The principle of the additional sunlight room is similar to that of the thermal insulation wall, which effectively blocks the direct contact between the hot indoor air and the cold outdoor air, thereby achieving a thermal insulation effect and reducing the energy demand of the building. Through software simulation, the heating time, heating efficiency, balanced temperature, building heat load reduction, and the time of using the new system throughout the year were calculated in detail. The specific results are as follows:Its working principle is not only to use the heat generated by cooking to heat the wall to raise the indoor temperature, but also to increase the heating effect by adding the characteristics of heat transfer between the sun room and the wall through solar radiation heating. The software simulation proved that the new system is effective for increasing the indoor temperature, and the temperature was increased by 4.16 °C.In the presence of solar radiation, the equilibrium temperature in the basic heating room was 6.5 °C, and the heating efficiency was 0.23 °C/h. After adopting the new system, the room temperature after equilibrium was 9.16 °C, and the heating efficiency was 4.09 °C/h.In the absence of WSR, the equilibrium temperature in the basic heating room was 5.5 °C, and the heating efficiency was 0.06 °C/h. After adopting the new system, the temperature in the room after equilibrium was 5.43 °C, and the heating efficiency was 2.87 °C/h. Therefore, when there was no WSR, the heating effect of the new system was obvious compared to the basic heating system.When the firewall and additional sunlight system were used, the thermal load of the building decreased by 1436.731 kW h during the whole year (with WSR). The thermal load of the building decreased by 735.919 kW· h throughout the year (without WSR). Taking 18 °C as the standard value of the building’s winter thermal comfort, the annual time below 18 °C was 3733 h (the wall has solar radiation) and 4006 h (the wall has no solar radiation).

### 4.2. Future Prospects

There are still some aspects of future research that need to be discussed and analyzed. The specific research content is as follows:According to actual investigations, the wall materials of the residential buildings in southern Shaanxi mainly involve clay porous brick masonry. Therefore, in the simulation analysis, the wall material of the model was set as clay porous brick masonry. As the maintenance structure of the building, walls are important for the insulation of the building. In future studies, the selection of wall materials can be an important research direction. Concrete, bamboo charcoal, or new types of materials can be used to find wall materials suitable for local conditions.The research aim of this paper was to put forward a firewall system with an additional sunlight room, focusing on the analysis of the impact of this system on improving the indoor temperature. It was verified that this system is effective for indoor heating in residential buildings. The ratio of walls to other elements is also an important factor affecting the heating of the system. In future research, this viewpoint can be used as the main research direction. First, the main elements affecting the indoor temperature can be determined based on the local geographical environment and climatic conditions to further determine the proportion of walls and major elements, so as to optimize the system.

## Figures and Tables

**Figure 1 ijerph-18-11147-f001:**
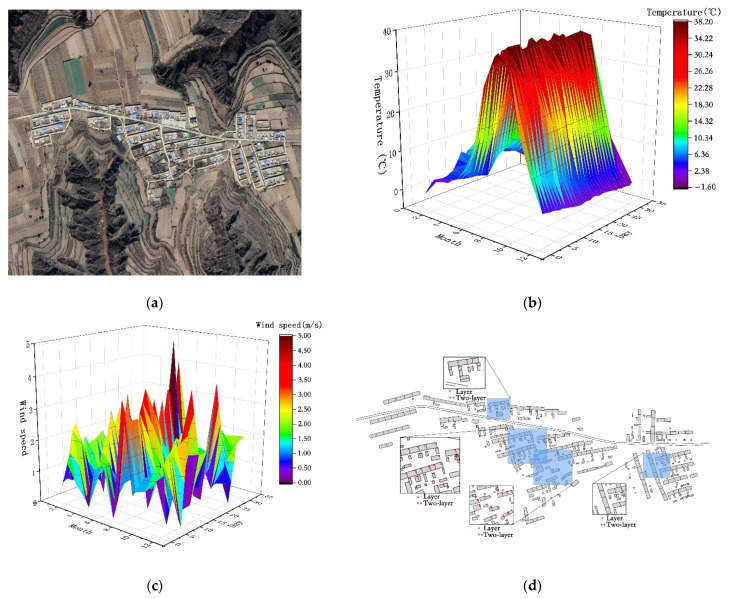
(**a**) Topographic satellite map of Hong village. (**b**) Annual temperature distribution map. (**c**) Distribution map of wind velocity throughout the year. (**d**) Village plan.

**Figure 2 ijerph-18-11147-f002:**
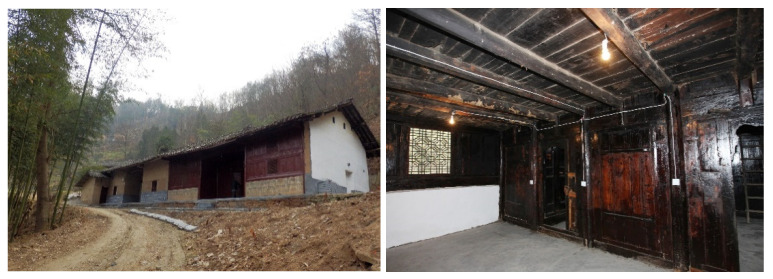
Photos of folk houses in southern Shaanxi.

**Figure 3 ijerph-18-11147-f003:**
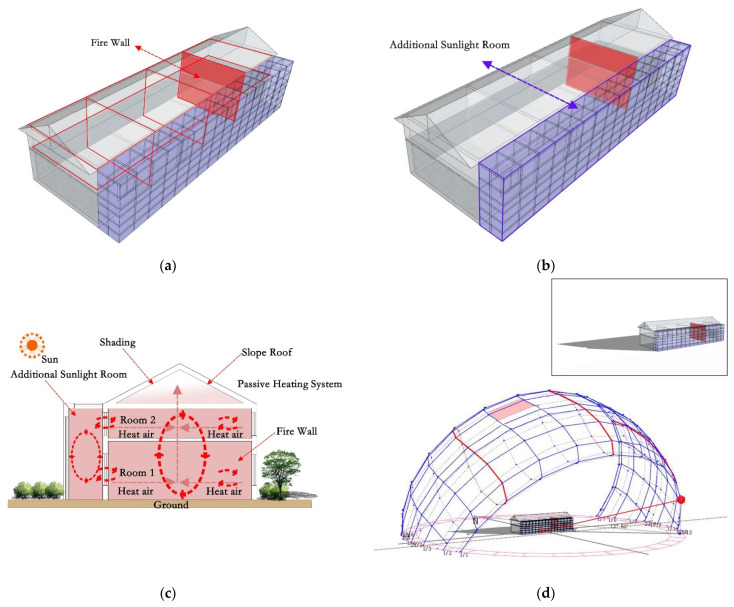
(**a**) The location of the firewall. (**b**) The location of additional sunlight. (**c**) Heat transfer effect picture. (**d**) Sunshine conditions in winter at 8:00 a.m. (**e**) Sunshine conditions in winter at 12 a.m. (**f**) Sunshine conditions in winter at 5:00 p.m. (**g**) Firewall system design in winter. (**h**) Firewall system design in summer.

**Figure 4 ijerph-18-11147-f004:**
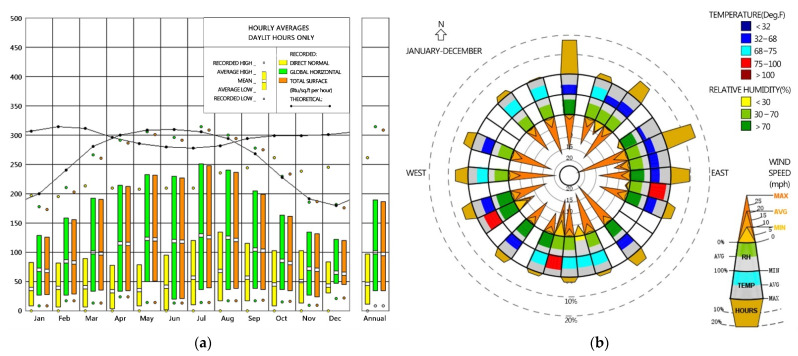
(**a**) Annual radiation map. (**b**) Annual temperature and humidity distribution map.

**Figure 5 ijerph-18-11147-f005:**
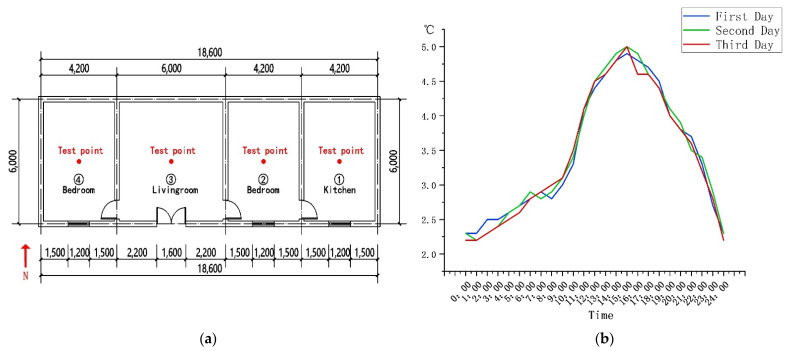
(**a**) Building layout. (**b**) Outdoor temperature distribution map. (**c**) Indoor temperature distribution map. (**d**) Outdoor temperature distribution during a day.

**Figure 6 ijerph-18-11147-f006:**
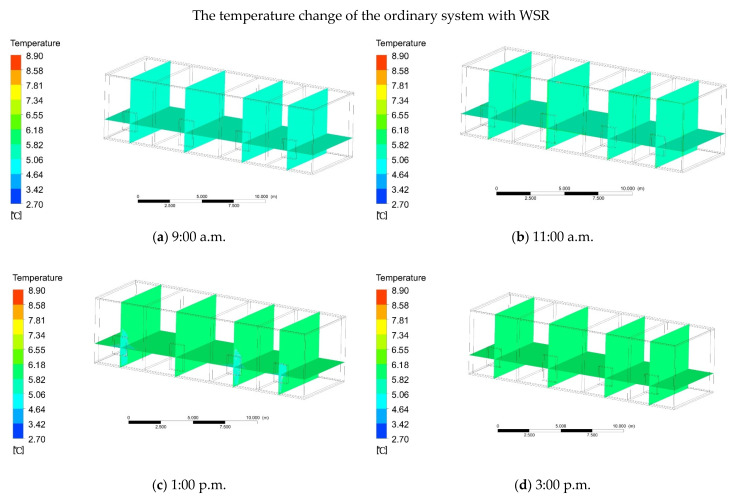
(**a**) At 9:00 a.m. the temperature of the ordinary heating (with WSR). (**b**) At 11:00 a.m. the temperature of the ordinary heating (with WSR). (**c**) At 1:00 p.m. the temperature of the ordinary heating (with WSR). (**d**) At 3:00 p.m. the temperature of the ordinary heating (with WSR). (**e**) At 5:00 p.m. the temperature of the ordinary heating (with WSR). (**f**) At 7:00 p.m. the temperature of the ordinary heating (withWSR).

**Figure 7 ijerph-18-11147-f007:**
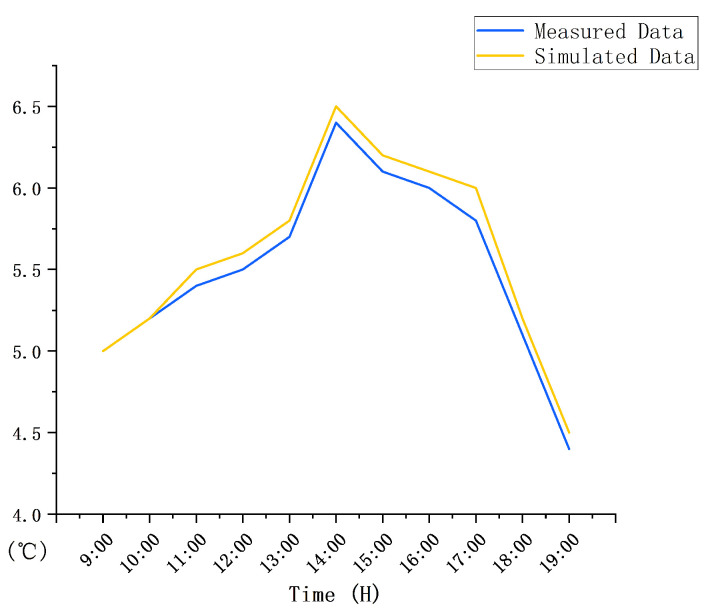
Comparison between the measured and simulated data.

**Figure 8 ijerph-18-11147-f008:**
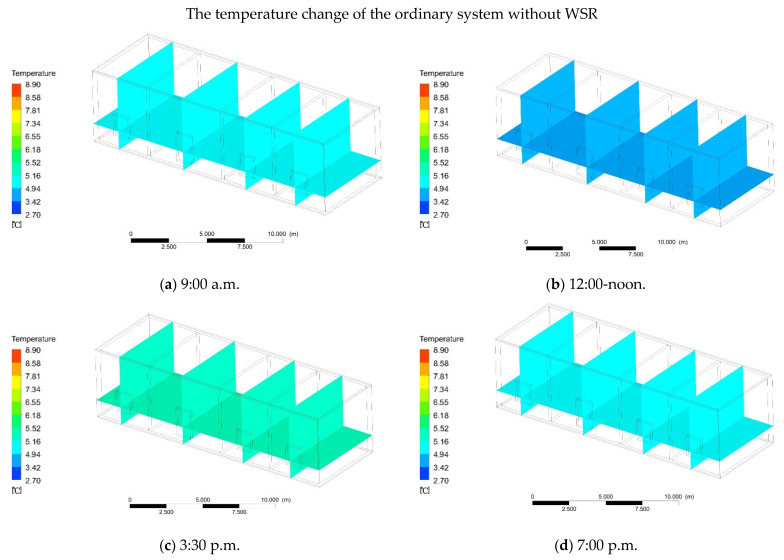
(**a**) At 9:00 a.m. the temperature of the ordinary heating (without WSR). (**b**) At 12:00-noon. the temperature of the ordinary heating (without WSR) (**c**) At 3:30 p.m. the temperature of the ordinary heating (without WSR). (**d**) At 7:00 p.m. the temperature of the ordinary heating (without WSR).

**Figure 9 ijerph-18-11147-f009:**
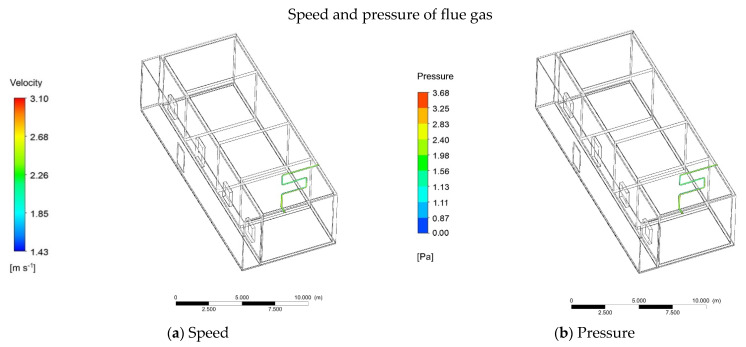
(**a**) Speed of the flue gas. (**b**) Pressure of the flue gas.

**Figure 10 ijerph-18-11147-f010:**
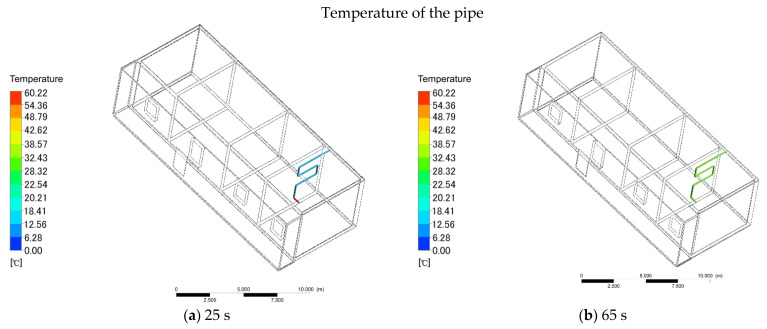
(**a**) Pipe heating in 25 s. (**d**) Pipe heating in 65 s. (**c**) Pipe heating in 95 s. (**d**) Pipe heating in 125 s.

**Figure 11 ijerph-18-11147-f011:**
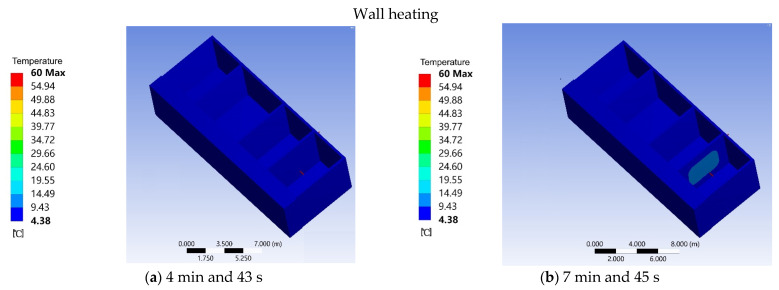
(**a**) Wall heating in 4 min and 43 s. (**b**) Wall heating in 7 min and 45 s. (**c**) Wall heating in 11 min and 30 s. (**d**) Wall heating in 16 min and 10 s. (**e**) Wall heating in 20 min and 10 s. (**f**) Wall heating in 24 min and 20 s.

**Figure 12 ijerph-18-11147-f012:**
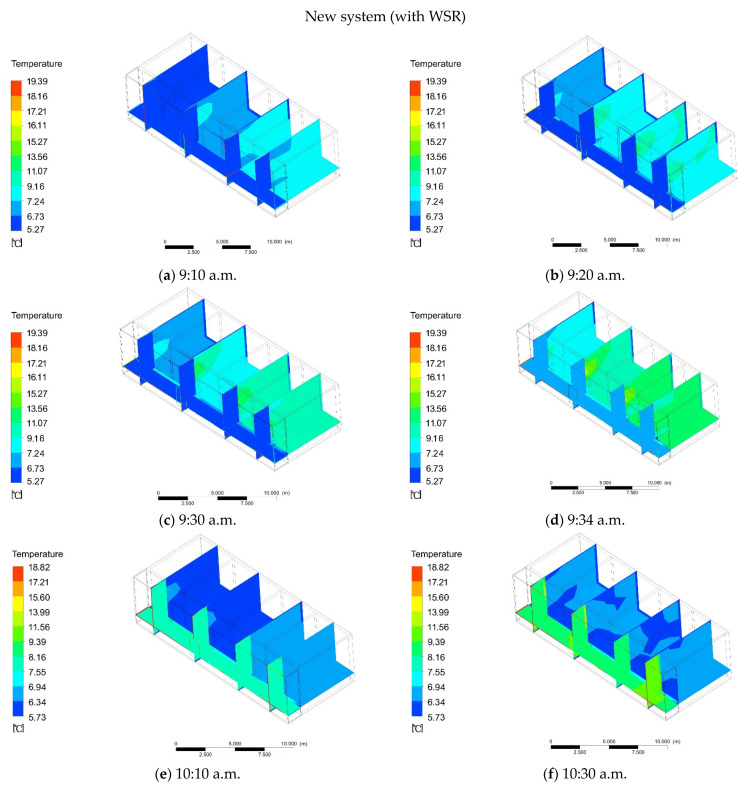
(**a**) At 9:10 a.m. the temperature of the new system (with WSR). (**b**) At 9:10 a.m. the temperature of the new system (with WSR). (**c**) At 9:30 a.m. the temperature of the new system (with WSR). (**d**) At 9:34 a.m. the temperature of the new system (with WSR). (**e**) At 10:10 a.m. the temperature of the new system (with WSR). (**f**) At 10:30 a.m. the temperature of the new system (with WSR). (**g**) At 10:50 a.m. the temperature of the new system (with WSR). (**h**) At 11:20 a.m. the temperature of the new system (with WSR). (**i**) At 11:36 a.m. the temperature of the new system (with WSR). (**j**)At 2:16 p.m. the temperature of the new system (with WSR). (**k**) At 6:00 p.m. the temperature of the new system (with WSR). (**l**) At 6:24 p.m. the temperature of the new system (with WSR). (**m**) At 6:45 p.m. the temperature of the new system (with WSR). (**n**) At 7:00 p.m. the temperature of the new system (with WSR).

**Figure 13 ijerph-18-11147-f013:**
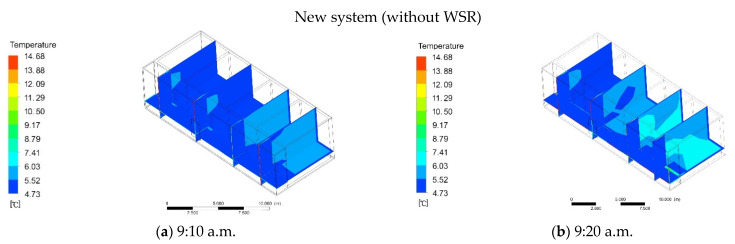
(**a**) At 9:10 a.m. the temperature of the new system (without WSR). (**b**) At 9:20 a.m. the temperature of the new system (without WSR). (**c**) At 9:30 a.m. the temperature of the new system (without WSR) (**d**) At 9:34 a.m. the temperature of the new system (without WSR). (**e**) At 9:45 a.m. the temperature of the new system (without WSR). (**f**) At 9:55 a.m. the temperature of the new system (without WSR). (**g**) At 10:10 a.m. the temperature of the new system (without WSR). (**h**) At 10:20 a.m. the temperature of the new system (without WSR). (**i**) At 6.00 p.m. the temperature of the new system (without WSR). (**j**) At 6.24 p.m. the temperature of the new system (without WSR). (**k**) At 6.45 p.m. the temperature of the new system (without WSR). (**l**) At 7.00 p.m. the temperature of the new system (without WSR).

**Figure 14 ijerph-18-11147-f014:**
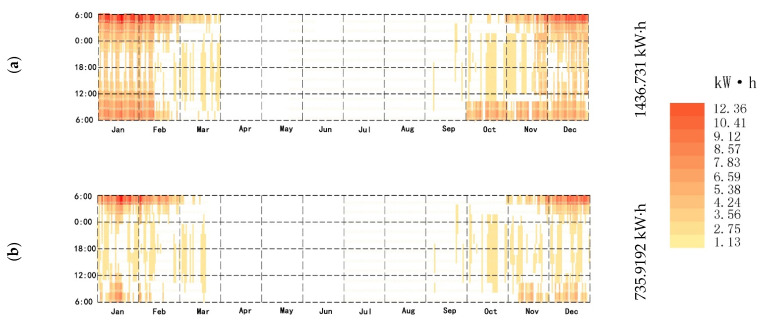
(**a**) Firewall–Sunlight system (with WSR). (**b**)Firewall–Sunlight system (without WSR).

**Figure 15 ijerph-18-11147-f015:**
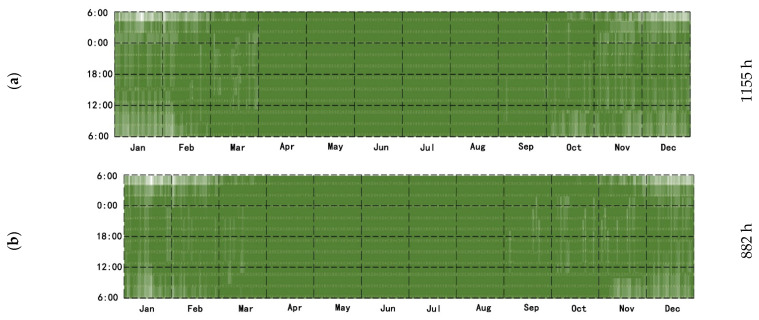
(**a**) Firewall–Sunlight system (with WSR). (**b**) Firewall-–Sunlight system (without WSR).

**Figure 16 ijerph-18-11147-f016:**
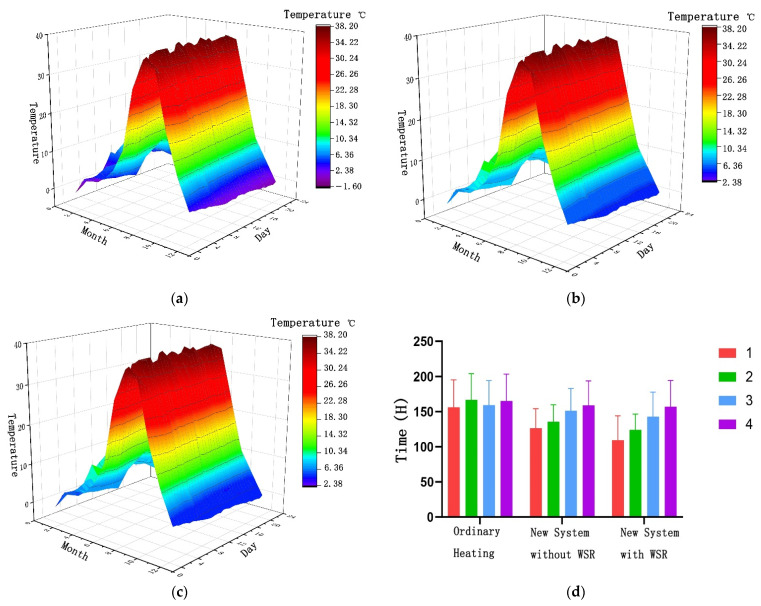
(**a**) Thermal comfort time under ordinary heating. (**b**) Thermal comfort time under the new system (with WSR). (**c**) Thermal comfort time under the new system (without WSR). (**d**) Thermal comfort time of the rooms.

**Figure 17 ijerph-18-11147-f017:**
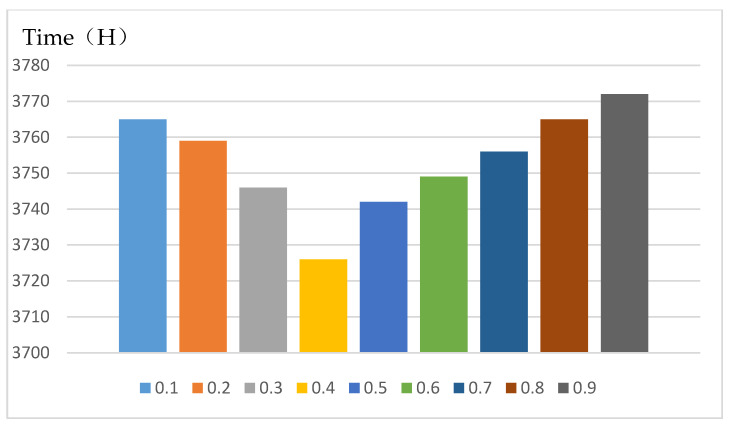
Optimization of the new system heating–window-to-wall ratio.

**Table 1 ijerph-18-11147-t001:** Basic information of the dwellings in southern Shaanxi.

Dwellings	Data Statistics
Research location	Hongcun, Houliu town, Shiquan county, Ankang city, and Shaanxi province
Climatic conditions	Hot summer and cold winter
Time and behavior	09:00–9:00 a.m. at home, 9:00–11:00 a.m. not at home, 11:00 a.m.–3:00 p.m. at home, 3:00–6:00 p.m. at home, 6:00 p.m.–12:00 a.m. at home
Number of occupants	1–2
Room information	Number: 4; sizes: Meeting room 6 m × 6 m, kitchen 4.2 m × 6 m, and bedroom 4.2 m × 6 m; height: 4.8 m
Roof	5 mm tile + 4 mm waterproof coiled material + 20 mm cement mortar + 4 mm lime cement mortar
Wall	4 mm lime cement mortar + 240 mm clay porous brick masonry + 3 mm cement mortar + 4 mm lime cement mortar
Window	Thickness: 6 mm; length: 1200 mm; width: 1300 mm; glass and heat transfer coefficient: 4.7 W/(m^2^·K); number: 3
Earthen stove	Length: 1800 mm; width: 900 mm; height: 800 mm
Door	45 mm wood; size: 900 mm × 2100 mm; number: 4
Floor	120 mm lime cement mortar
Stoves	Length: 1800 mm; width: 900 mm; height: 800 mm

**Table 2 ijerph-18-11147-t002:** Related properties of the building materials.

Material Name	Thermal Storage Coefficient W/(m^2^·K)	Specific Heat Capacity (J/kg·K)	Density (kg/m^3^)	Thermal Conductivity (W/m·K)
Tile	6.23	1406	1800	0.43
Waterproof coiled materials	3.302	1470	600	0.17
Cement mortar	11.37	1050	1800	0.93
Lime cement mortar	10.75	1050	1700	0.87
Clay porous brick masonry	6.602	1356	850	0.52
Wood	3.575	2510	500	0.14
Air	-	-	1.27	-
Aluminum	191.495	920	2700	203

Note: Data from the appendix of *Building Physics*.

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
