# Peer review of "Study on Passive Heating Involving Firewalls with an Additional Sunlight Room in Rural Residential Buildings"

_ijerph, 2021, doi:10.3390/ijerph182111147_

Round 1

Reviewer 1 Report

The manuscript entitled “Study of passive heating in fire wall-additional sunlight room of rural residential buildings” deals with an approach of passive heating named heat recovery with solar-assisted passive heating. Results of simulation show that the hours of thermal discomfort (set hereby at 18 °C) decrease.

The manuscript has to be improved in a deeper manner for being considered as suitable for publication. The most concerning points are stated as follows:

  • Results must be validated. The simulation model lacks in much input data such as the climate conditions, schedule and number of the occupants, and physical characteristics of the materials of construction, to name only a few. In the present form, results can be easily questioned and neglected.
  • There is not a real comparison between rural and urban dwellings in terms of architectural design, climate conditions and occupants’ behavior; even though authors claim that the manuscript is novel in its analysis.
  • The manuscript is very hard to follow due to the grammar mistakes, but especially by the fact that many of the descriptions might be set with a scheme or a picture, such as the situation of the description of the fire-wall approach.
  • Due to the scope of the journal, the discussion regarding thermal comfort must be much deeper, instead of the five lines written in the manuscript.
  • A typical meteorological year with at least outdoor temperature, relative humidity and global solar radiation is necessary to fully understand the context of the simulations.

Specific comments:

Line 9: Describe TCE.

Line 29: The keywords are sort of ambiguous and they do not provide a clear description of the subject.

Line 243: The data collection is mentioned but is not show within the manuscript.

Lines 338, 343 & 353: Please set the source of Eq. 1-3.

Reviewer 2 Report

The paper presents the results for the thermal response of buildings considering new passive heating system. The detailed comments are as follows:

  1. The dimension of house models studied should be added.
  2. The detailed measurement system (device, number, arrangement (location and direction), duration of instrumentation) should be added.
  3. The results of Ordinary Heating (with WSR); New System (with WSR); Ordinary Heating (without WSR) are shown. However, the order of the three aspects should be given: OH(w/o WSR), OH(w WSR) and NS (w WSR). Additionally, OH and NS are not matchable because ordinary heating means “heating” and new system is not “heating” but “installation”. Consider proper terminology and clarify the groups.
  4. The information related to ANSYS model is totally missed in main text: geometry and material properties of the model, modelling mesh, interface conditions and boundary conditions
  5. Compare the measured and computed(simulated) results of the models. This is related to verification of this model study.
  6. The size and resolutions of figures are too low to understand the numerical values as well as to compare the results. They should be improved.

Reviewer 3 Report

  • The article needs major grammatical and syntax improvements. Use of English service center is recommended. 
  • The results mentioned din abstract should be indicate more of percentages of improvement.
  • It would much more effective that models with fire wall and without fire walls are considered and compared at the same condition, to show the efficiency of the new system. This can be conducted with closed greenhouse and without it.
  • The author mentioned “some ex- parts have proposed the use of renewable sources, such as geothermal, hydroelectric power generation and other methods to save energy’ Following references are suggested to be considered for indication of ways to save energy by various procedures.
    • Experimental investigation of sound transmission loss in concrete
    • Nano silica and metakaolin effects on the behavior of concrete containing rubber crumbs. CivilEng, 1(3), 264-274.
    • Temperature and humidity effects on behavior of grouts. Advances in concrete construction, 5(6), 659.
    • Investigation of steel fiber effects on concrete abrasion resistance. Advances in concrete construction, 9(4), 367-374.
    • Compressive behavior of concrete under environmental effects. In Compressive Strength of Concrete. IntechOpen.
  • The total height is 4.8 m or 4.5m for the buildings since the height of the first floor is 3.3m and the second floor is 1.5m
  • For figure 3 f more explanation and details are required.
  • How is the simulation procedure verified?
  • Is there a way to actively control the heat transfer from the fire wall for once the temperature drops.
  • The ratio of the wall to other elements could be considerably change the results, would be more effective if the results normalized based on the wall surface area.

Round 2

Reviewer 1 Report

All my comments were correctly addressed. I commend the authors for the effort. Now I can claim that the manuscript is suitable for publication.

Author Response

Reviewer 1 did not propose modification opinions in the  round 2.

Reviewer 2 Report

The reviewer has examined the revision made by the authors. It is shown that that the comments involving the measurement system, modeling details and model verification are reflected in the revised manuscript. However, for comparision between results of modelling and measurement should be shown in figures (currently stated only in text).

Reviewer 3 Report

Some normalized results should be shown. The ratio of the wall to other elements could be considerably change the results, would be more effective if the results normalized based on the wall surface area.

For the following new statements added "In future studies, we should consider how to actively control the heat transfer of the firewall when the temperature drops. The heat transfer of the firewall depends on the insulation and heat storage performance of the material, thereby saving energy consumption. Common insulation materials include polystyrene boards, foam cement boards, and other materials. These materials can be combined with the firewall, and suitable heat preservation and heat storage materials can be selected through experiments and simulations.:  The following works are recommended to be considered to complete reference :

[1]Experimental investigation of sound transmission loss in concrete.
Nano silica and metakaolin effects on the behavior of concrete containing rubber crumbs. CivilEng, 1(3), 264-274.[2]Compressive behavior of concrete under environmental effects. In Compressive Strength of Concrete. IntechOpen.  [3]Nano silica and metakaolin effects on the behavior of concrete containing rubber crumbs. CivilEng, 1(3), 264-274. [4]Temperature and humidity effects on behavior of grouts. Advances in concrete  construction, 5(6), 659.[5]Investigation of steel fiber effects on concrete abrasion resistance. Advances in concrete construction, 9(4), 367-374.

The shortcomings from costs to installation should be elaborated
